# Measuring single-cell susceptibility to antibiotics within monoclonal bacterial populations

**Lena Le Quellec**[1,2☯], **Andrey Aristov**[1☯], **Salomé Gutiérrez Ramos**[1,2], **Gabriel Amselem**[1,2], **Julia Bos**[3], **Zeynep Baharoglu**[3], **Didier Mazel**[3], **Charles N. Baroud**[1,2]*

1 Institut Pasteur, Université Paris Cité, Physical Microfluidics and Bioengineering, Paris, France, 2 LadHyX, CNRS, Ecole Polytechnique, Institut Polytechnique de Paris, Palaiseau, France, 3 Institut Pasteur, Université Paris Cité, CNRS UMR3525, Bacterial Genome Plasticity Unit, Paris, France

☯ These authors contributed equally to this work.
* charles.baroud@pasteur.fr

**Data Availability Statement:** Processed data is included inside the manuscript and its Supporting information files. Raw data is freely available on Zenodo https://zenodo.org/records/6940212 and

## Abstract

The emergence of new resistant bacterial strains is a worldwide challenge. A resistant bacterial population can emerge from a single cell that acquires resistance or persistence. Hence, new ways of tackling the mechanism of antibiotic response, such as single cell studies are required. It is necessary to see what happens at the single cell level, in order to understand what happens at the population level. To date, linking the heterogeneity of single-cell susceptibility to the population-scale response to antibiotics remains challenging due to the trade-offs between the resolution and the field of view. Here we present a platform that measures the ability of individual *E. coli* cells to form small colonies at different ciprofloxacin concentrations, by using anchored microfluidic drops and an image and data analysis pipelines. The microfluidic results are benchmarked against classical microbiology measurements of antibiotic susceptibility, showing an agreement between the pooled microfluidic chip and replated bulk measurements. Further, the experimental likelihood of a single cell to form a colony is used to provide a probabilistic antibiotic susceptibility curve. In addition to the probabilistic viewpoint, the microfluidic format enables the characterization of morphological features over time for a large number of individual cells. This pipeline can be used to compare the response of different bacterial strains to antibiotics with different action mechanisms.

## Introduction

Antimicrobial resistance is considered by the World Health Organization as one of the biggest threats to public health [1, 2]. Developing new tools and methods to better understand bacterial resistance is becoming necessary. When addressing the question of bacterial response to antibiotics, most microbiology studies report the Minimum Inhibitory Concentration (MIC) at which the cells stop growing, for a given initial inoculum size (usually $10^5$ cells) and after a growth time of over 24 hours. However, interpreting these MIC measurements is far from

the code is available at https://github.com/
BaroudLab/anchor-droplet-chip.

**Funding:** ANR grant UniBAC (ANR-17-CE13-0010).

**Competing interests:** The authors have declared
that no competing interests exist.

trivial and could indicate many different phenomena taking place in the cultures (high variability, inoculum effect, mechanism of action of the antibiotic) [3]. Indeed, the response of individual cells, which may then lead to colonies, can display large heterogeneity [4, 5]. For this reason the MIC is sometimes complemented with a more precise measurement, the Minimum Bactericidal Concentration (MBC). The process of determining the MBC is heavy and time-consuming and, like the MIC, only gives information about the antibiotic susceptibility on the population level. This has motivated significant effort to measure the bacterial response at a more microscopic level.

For instance, a single-cell MIC was estimated by measuring the MIC for different inoculum sizes and extrapolating the value to a single cell [6]. Classical laboratory methods, however, are difficult to scale to single-cell manipulation, both in terms of the volumes of interest and also the number of experiments that are required to obtain a significant number of replicates. In contrast, the development of microfabrication methods and microfluidics has allowed measurements to be made on individual cells in controlled environments [7]. As a result the microfluidics literature contains a wide range of designs to test bacterial response to antibiotics (see e.g. [8–11]).

In particular two dominant platforms have emerged that have been developed by several labs: First the so-called mother machine and its variations, where individual cells are trapped in thin channels and observed over a large number of generations [12]. These devices rely on the tracking of the initial *mother* cells by time-lapse microscopy, while removing its daughter cells as they push out of the microchannels. This method has been used for rapid detection of antibiotic susceptibility for a range of different antibiotics [13]. Moreover by fine analysis of the images under different antibiotic treatments it is possible to learn about relations between mother and daughter cells [14], or to detect the effect of rare mutations on the fitness at the single-cell level [15].

In parallel, the field of droplet microfluidics has allowed studies of a different kind. By encapsulating one or a few bacterial cells within water-in-oil droplets, the development of small colonies from individual cells [16], or the signature of their metabolism [17], can be detected with optical readouts. The addition of antibiotics in solution within the droplets can then be used to determine the bacteria's susceptibility. This basic principle has, in recent years, been developed in two main directions. Either to improve the simplicity of use [18, 19] or to improve the precision of the measurements [20–22]. These droplets approaches have the potential to be transferred to clinical studies, as reviewed in recent papers [23–25].

Although these droplet methods constitute important milestones, they suffer from several drawbacks: First, they require specific and sophisticated equipment including precise flow control systems to ensure droplet size homogeneity, as well as high-speed electronics, lasers, and data acquisition, to perform the measurements on flowing droplets. Second, the latest methods do not allow the droplets to be followed in time or to relate the final state to the initial state of the droplets. Finally, there is still a need to strengthen the link between the droplet-based measurements with the vast quantity of data obtained in traditional experiments.

Here, we present an open-access microfluidic platform that addresses some of these issues. The platform is based on *rails and anchors* that were introduced a few years ago [26]. Those droplets are formed within microfabricated wells and remain stationary for the duration of the experiment, including for the observation of biological processes within them [27, 28]. As such, the platform only requires simple microfabrication, low-precision flow control and allows to complete time-lapse measurements. The microfluidic setup is augmented with an original and dedicated image acquisition and analysis pipeline that extracts the relevant information from the chips in an automated manner. By providing the image and data analysis as open-source code, the platform will be easy to integrate in most academic laboratories.

Moreover, the current study addresses the interpretation of the biological measurements for the first time. By doing so it links the droplet-level approach, used in most droplet-based experiments, with a single-cell analysis. This analysis is used to obtain unique measurements of the single-cell susceptibility to antibiotics.

## Results

### Microfluidic platform and initial observations

The microfluidic device used in this study is based on a geometry described previously [28, 29]. It consists of a triangular array of 501 individual wells, or anchors, that can each hold a single aqueous droplet (Fig 1a). The chip format is well suited for imaging on an inverted microscope, either using wide-field illumination or confocal mode, as discussed below.

Loading the chip and distributing the bacterial solution into droplets is straightforward and takes about 5 min (see Method for the detailed steps, and Fig 1b–1d). Briefly, the device is first entirely filled with oil (FC40, 3M) containing 0.5% surfactant (FluoSurf, Emulseo, France). Then, the aqueous suspension of bacteria is flowed through the device. In a last step, the oil is flowed again through the device, which leads to the formation and immobilization of droplets containing bacteria, directly on the anchors. The volume of each droplet is 2 nl and is primarily determined by the volume of the anchor [27]. This loading procedure is simple to set up and can be mastered in only a couple of trials. Indeed, the droplet formation step is robust to fluctuations in flow rate. It can be performed using syringe pumps or using hand-held syringes. Also note that the whole loading procedure is performed while the output is connected to a waste tube, and the bacterial suspension is constantly encapsulated in an inert oil layer. This makes the protocol suitable in principle to handling pathogens, since the bacterial solution remains in a closed circuit.

*E. coli* W3110 with a fluorescent reporter (RFP) in the ptac site (ptac::RFP) allows us to detect bacterial cells and the growth of the colonies using the red fluorescent protein (RFP). A sample fluorescence image of the trapping area of the chip, acquired after 24 hours of incubation, is shown in Fig 1e. The bacteria at this stage form bright fluorescent colonies that can readily be imaged using epifluorescence or confocal microscopy. Empty droplets are also present on the device, since the cells are randomly distributed. The number of empty droplets is related to the average number of bacteria per drop, which is in turn related to the initial concentration of the bacterial suspension. Starting from different initial dilutions, therefore, allows us to tune the average number of cells per droplet and the number of positive droplets within the chip in the absence of antibiotics.

Before moving on to testing the effects of antibiotics in the microfluidic device, we benchmark the bacterial fitness in the microfluidic device compared with standard multiwell plate experiment. This is done by following the growth of the fluorescent intensity in the droplets, using time-lapse epifluorescence microscopy, while performing in parallel a standard growth-curve measurement on a fluorescence plate reader from the same batch culture. The growth curves for the two cases are shown in Fig 1f. The curves from the microfluidic device show a large variability between individual droplets, as a result of distribution of the number of initial cells and of the cell-to-cell variability [30]. In contrast, the multiwell plate experiments are insensitive to these stochastic elements and grow in a reproducible and regular manner. Fitting the individual growth curves with an exponential function for the first 10 hours of growth shows that the difference in growth rates in the microfluidic device and in the multiwell plates is statistically not significant, see Fig 1g. Hence, the microfluidic results can be compared to standard microbiology techniques.

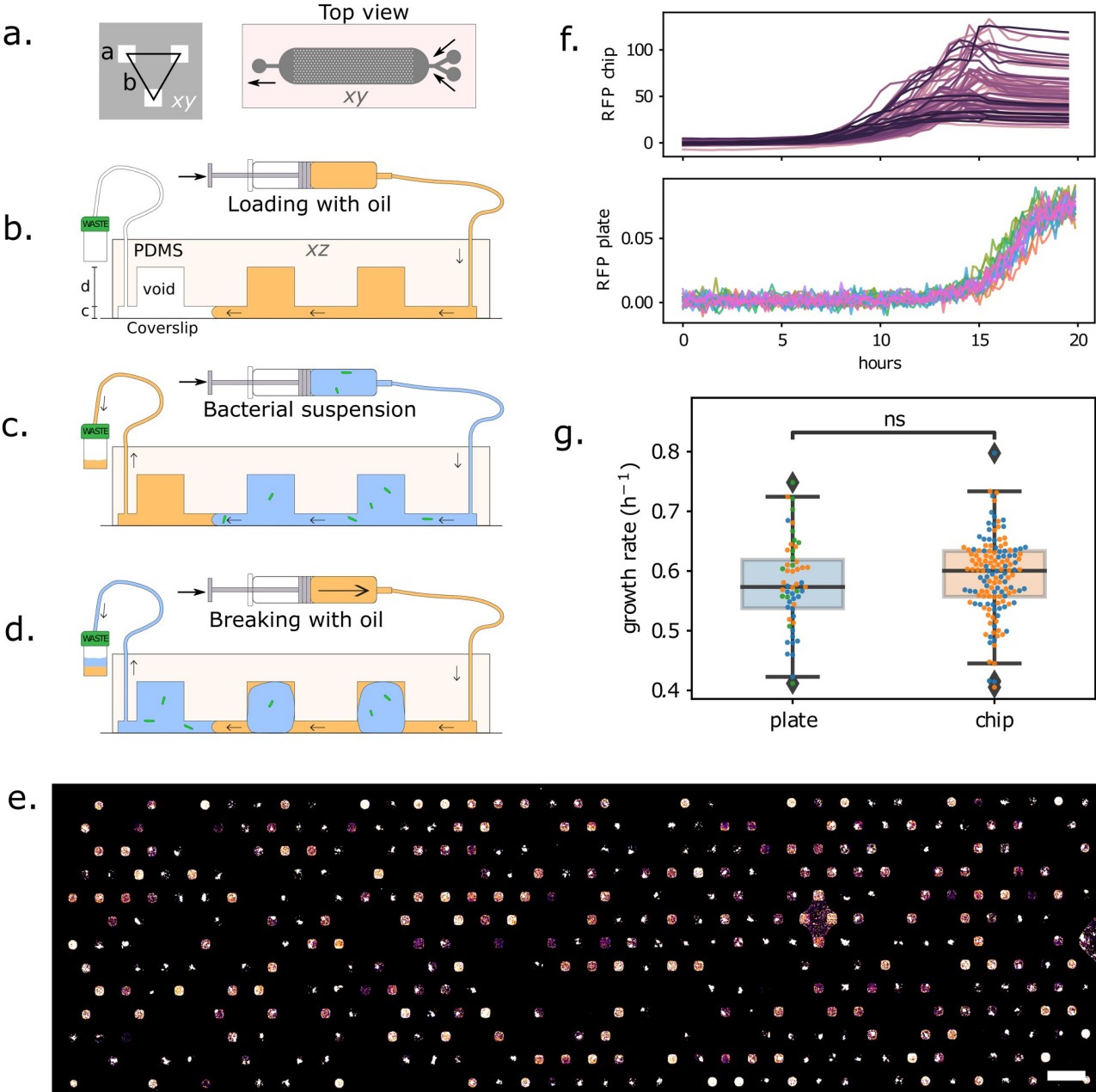

**Fig 1.** **(a)** Microfluidic chip design. Left: a unit cell consists of a triangular arrangement of square anchors, each of size $a$ = 120 μm and distance $b$ = 360 μm. Right: design of the full chip, which includes a main chamber with 501 anchors, 2 input channels on the right and 1 outlet channel on the left. **(b)** Side-view (not to scale), showing the channel height $c$ = 30 μm and anchor depth $d$ = 100 μm. The loading protocol begins by filling the chamber with oil, then **(c)** replacing the oil with bacterial suspension, and **(d)** breaking the bacterial suspension into individual droplets, anchored in their respective wells. **(e)** Z-projection of a confocal stack of the chip after 24 h incubation (scalebar: 500Â μm). **(f)** Growth curves in 501 individual droplets on the chip (top), and in each well of a 96-well plate (bottom). **(g)** Measured growth rates for bacteria in the microfluidics chip (2 replicates), and in the 96-well plate (3 replicates). Growth rates were obtained by fitting an exponential function to the growth curves during the first 10 hours of growth. P-value = 0.24 obtained with Welch's t-test for independent samples.

## Imaging and analysis pipelines

Once the microfluidic device is loaded, the aim of the experiments is to identify which droplets within the array produce a population of cells after 24h and to link the final state with the initial number of cells in each droplet. These measurements are performed by first imaging the chip shortly after the loading and then after overnight incubation. Image analysis of the initial and final time points then yields a table that identifies each droplet in the array. Each droplet is then associated with measured quantities such as the initial number of cells, its final state, as well as the antibiotic concentration for a given experiment (see schematic in Fig 2).

As such, the ability to acquire and analyze large amounts of imaging data is fundamental to obtain the antibiotic response curves. While the array format lends itself naturally to measuring droplet contents at different time points, the requirement to detect single cells at early times imposes high-resolution imaging. This runs into data handling limitations associated with the large file sizes and large number of experiments. These different constraints led us to develop automated imaging and analysis pipelines whose implementation was instrumental for obtaining the results below. The imaging steps are described below and the image and data

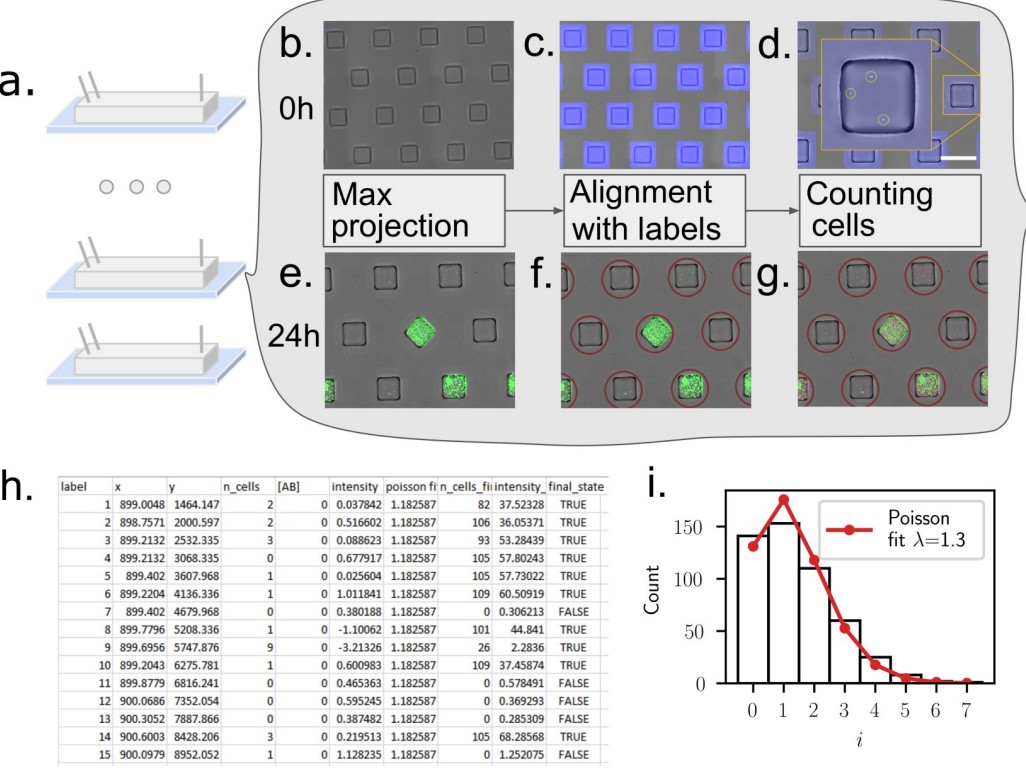

**Fig 2. Imaging and image analysis pipeline. (a)** The experiment begins by loading several chips with bacterial suspensions at different antibiotic concentrations. **(b)** Overlay of the bright-field image of the chip, and of the maximum projection of a fluorescent confocal z-stack. Images are acquired minutes after loading (0 h). **(c)** The bright-field image is used to identify the alignment with respect to the microscope stage and to create a unique mask around the positions of the individual anchors. **(d)** Detecting the fluorescent peaks in the maximum projection enables to count the initial number of cells per droplet. **(e)**-**(g)** Similar imaging and analysis operations are performed 24 h after the loading, to identify positive and negative droplets. **(h)** The data from the initial and final images are collected in a table that provides a unique label for each droplet, as well as the number of cells at 0 h and at 24 h. Each droplet is assigned a final status of positive or negative, depending on whether the final number of cells is larger than a predefined threshold of 15 cells. **(i)** A quality control step is performed using the single-cell detection at $t = 0$ by comparing the cell number distribution with a Poisson distribution. This leads to an estimate of the Poisson parameter λ for each chip.

analysis pipelines are provided as open-source packages at the following GitHub repository: https://github.com/BaroudLab/anchor-droplet-chip.

The aim of the pipeline is to generate a table with one line for each droplet, which includes a unique label, the antibiotic concentration, as well as the initial number of cells and final state for the droplet, as described in Fig 2.

**Initial state.** The initial number of cells per droplet is obtained by imaging the freshly loaded chip first in bright-field, and second by acquiring a confocal z-stack of the device in the RFP channel. The bright-field image is used to detect the positions of the droplets, by adjusting for arbitrary shift and tilt of each acquisition (see Methods). In turn, the confocal stack is reduced to a single image by using a maximum projection of the fluorescent intensity. The resulting fluorescent image allows us to count the number of individual bacteria within each droplet (Fig 2c). The algorithm proved sufficiently robust to perform unsupervised automated analysis of the data from the chips.

Since the cells are expected to follow a Poisson distribution in the droplets [28], a quick quality control is performed at the end of the first scan, by verifying the distribution of number of cells per droplet, and checking that it indeed follows the expected shape, see Fig 2i. This calculation also allows us to obtain a value of the Poisson parameter $\lambda$ and to adjust the cell dilution if necessary in order to work in the desired range of $\lambda$.

Note that the loading and scanning each take about 10 minutes and the initial image validation occurs within a few minutes as well. This yields a first measurement of the loading in under 30 minutes for each microfluidic chip.

**Final state.** The contents of each droplet are later measured at the experimental endpoint, typically after an overnight incubation ($t = 18 - 24$ h): each chip is scanned, this time using simple epi-fluorescence to optimize time and disk storage. The final image of each chip is re-aligned with the template acquired at initial times to identify each droplet, a registration step made straightforward by the fact that droplets are anchored at predefined positions. Then, cells are detected within each droplet, as shown in Fig 2e–2g. In the current experiments, we focus on the bottom of the microfluidic device where cells are more likely to be detected if the droplet is positive. Different quantities can be obtained from the final image as proxies for the ability of cells to grow within the drop. We count the number of cells in the final image, although mean or total fluorescence intensity can also be used. The main requirement for the measurements is to be sufficiently robust to yield a cutoff between positive and negative droplets, a classification which is also included in the data table. The final result is a csv table that contains the relevant information on the initial and final states of each droplet within each microfluidic device.

Note that the protocols described above can be modified for particular situations. For instance time-lapse microscopy can be performed on the chips to obtain time-resolved measurements. Similarly, confocal imaging can be used at later times to obtain a more precise cell count or fluorescent intensity, or to identify the morphology of the cells. Although these cases would require small modification in the pipeline, the main bricks of the analysis discussed above can still be used in a modular fashion.

## Microfluidic *vs.* microplate antibiogram

The microfluidics and imaging protocols described above can be combined to obtain an antibiotic susceptibility curve, by loading several chips in parallel, using known concentrations of antibiotics and bacteria in each chip. Performing these measurements is simplified by the standardized microfluidic format and analysis codes, making it possible to run six to twelve chips in parallel, each with a different concentration. Confronting the microfluidic measurements

against standard microbiological techniques proves crucial to understand how to interpret the microfluidics data.

In each experimental run, one chip is loaded with the same bacterial concentration as the others but without any antibiotic. This control chip allows us to estimate the value of the average number of cells per droplet at initial times, λ, for the given run:

$$\lambda \simeq -\ln(\hat{p}_-), \tag{1}$$

where $\hat{p}_-$ is the fraction of empty to total droplets on the test chip. This estimate comes from the assumption that the initial number of cells per droplet follows a Poisson distribution [30]. A higher value of λ means that drops contain a larger average number of cells initially.

A typical set of images from the experimental endpoint is shown in Fig 3a. In these images the bright spots correspond to droplets where bacteria grew, while dark spots within the regular matrix correspond to droplets that do not contain a sufficient population of cells. These

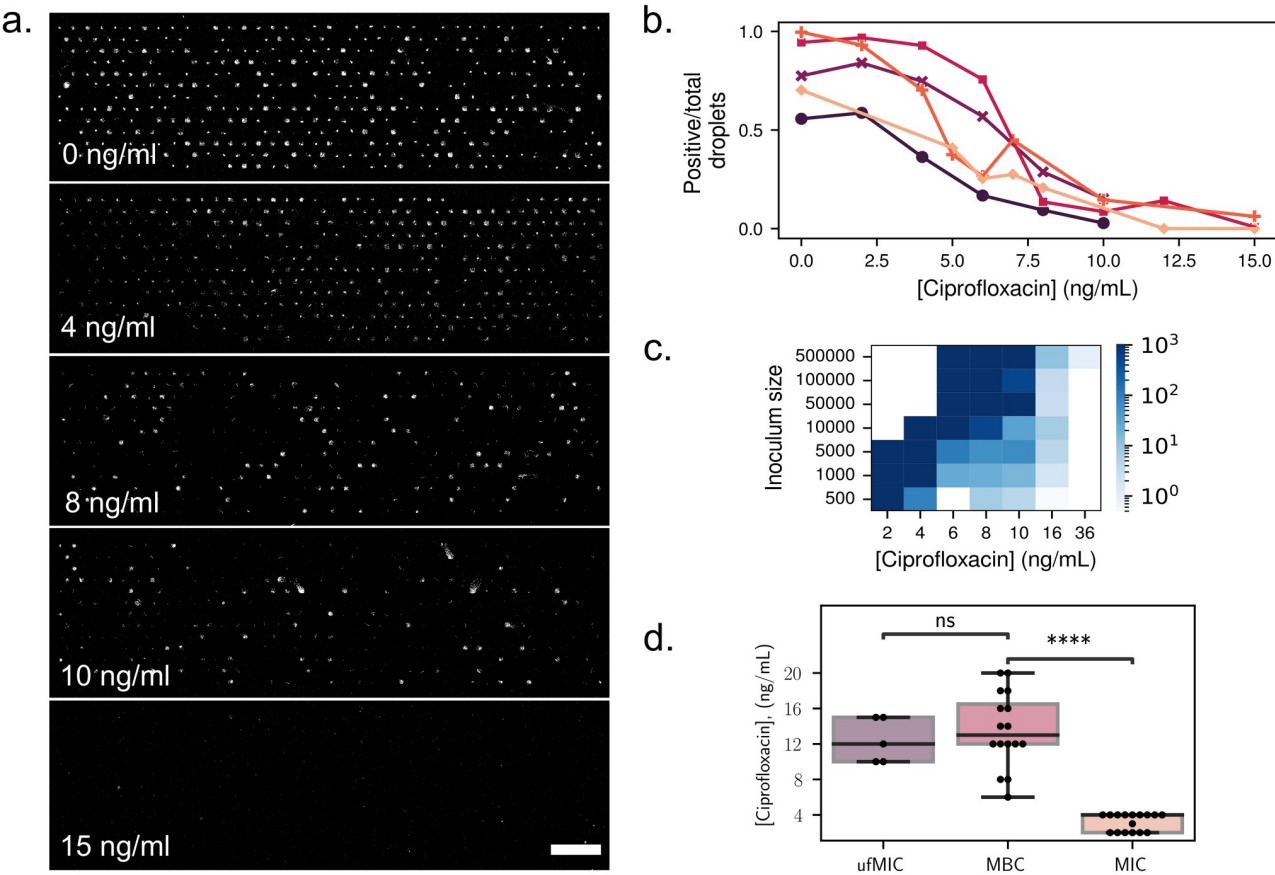

**Fig 3.** **(a)** Antibiogram chips with different concentrations of ciprofloxacin, and an initial average number of cells per droplet λ = 1.3. Bright spots correspond to droplets where bacterial growth occurred. Scale bar: 1 mm. Fraction of positive droplets at the end of the experiment, normalized by the total number of droplets containing at least one bacterium at the beginning of the experiment, for different concentrations of ciprofloxacin. Each color corresponds to a different run, at a different date and with a different value of λ. **(b)** Ratio of positive to negative droplets for different experimental replicates, corresponding to different values of λ. **(c)** Result of MBC experiments. Bacteria were grown in 96-well plates under different antibiotic concentrations, and starting with different inoculum sizes. The wells were scanned after 24 hours, and the contents of negative wells were replated on antibiotic-free petri dishes for 24 additional hours. The number of colonies on the petri dishes after incubation directly represents the number of surviving bacteria in each well. The color corresponds to the number of colonies on the petri dishes. **(d)** Values of the MBC (petri dishes), MIC (96-well plates) and μf · MIC on the chip, where the total number of cells per chip is used as inoculum size. P-values = 6e-8 for MBC vs. MIC and 0.9 for μf · MIC vs MBC obtained with Welch's t-test for independent samples.

dark positions correspond either to droplets that did not contain any cells initially or to droplets where the cells did not form colonies e.g. due to the antibiotic stress. This antibiogram, allows to determine a measure of the antibiotic susceptibility of the bacteria, which we denote µf · MIC. The µf · MICcorresponds to the lowest antibiotic concentration that will inhibits the growth in 95% of the droplets.

As expected, the fraction of positive droplets decreases as the concentration of antibiotics is increased. This decrease is quantified in Fig 3b, where the number of positive droplets is shown to decrease towards zero as the concentration of the antibiotic ciprofloxacin increases, independently of the value of λ.

The interpretation of this "digital" measurement and its relation with classical microbiology measurements is not obvious. To understand its significance, measurements from the microfluidic format were confronted with measurements in a standard multiwell plate, using the same samples. Two classical microbiology measurements were performed on the samples. First, the minimum inhibitory concentration (MIC) was determined with a standard microtitre broth dilution method (see Methods). The MIC corresponds to the lowest antibiotic concentration that still inhibits visible growth. Then, the minimum bactericidal concentration (MBC) was obtained by replating the contents of the negative wells on antibiotic-free petri dishes. The number of colonies on the petri dishes after 24 hours of incubation directly represents the number of surviving bacteria in each well. The MBC thus provides a more precise measure of the antibiotic concentration that is lethal to the bacteria, compared to the MIC. We find that the number of colonies growing from negative wells is very large below a critical value of the ciprofloxacin concentration of 10 ng/mL, after which it drops dramatically to below ≈100 colonies, and eventually asymptotes to zero with increasing antibiotic concentration, see Fig 3c.

A comparison of the measurements obtained from the three techniques (µf · MIC, MIC, and MBC) is shown in Fig 3d, as a function of inoculum size. The inoculum size used for the µf · MIC corresponds to the total number of cells per chip. The classical MIC measurement (at $t = 24$ h) is shown to be the least sensitive of the three measurements, since it finds a critical value of the concentration that is well below the value that is necessary to kill all of the cells. In contrast, the values of the MBC and of the µf · MIC are in the same range. They both show similar trends with the inoculum size, namely a slow but detectable increase with the initial number of bacterial cells.

This close correspondence between the MBC and the µf · MIC, obtained by pooling together the total number of cells in the chip, indicates that the results in the microfluidic chip can be treated as a population-level measurement: all the bacteria from a single chip form a small population, whose ability to survive to a given concentration of antibiotics depends on its initial size. This result indicates that cells are behaving in an independent manner.

## Computing single-cell antibiotic susceptibility

Beyond population level measurements, the objective is to provide insights about the antibiotic susceptibility of individual cells within a monoclonal population. This is achieved by taking a probabilistic viewpoint on the ability for a single cell to produce a colony at a given antibiotic concentration. Experimentally, we count the number of bacteria in each droplet at the beginning and at the end of an experiment for six antibiotic concentrations, see Fig 4a. At the end of an experiment, the image analysis algorithm counts the number of bacteria at the bottom of the droplets. While this number does not reflect the exact total number of cells in the droplet at late times, it is an acceptable proxy to differentiate positive from negative droplets.

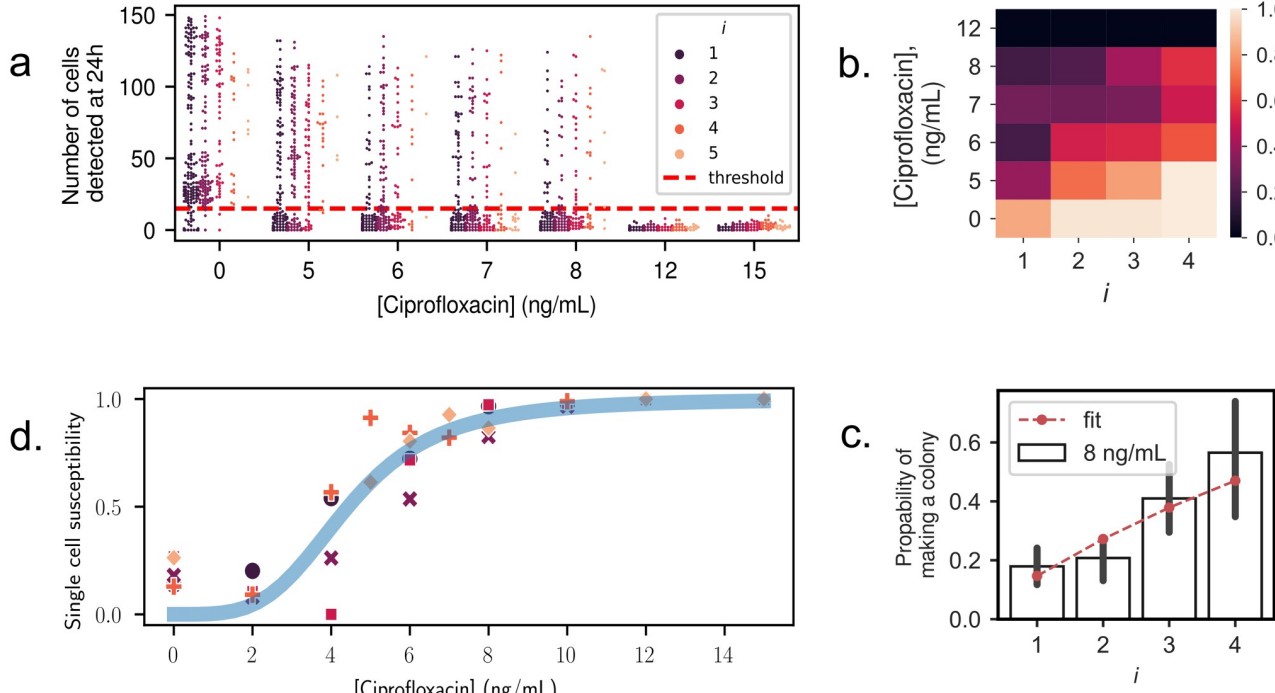

**Fig 4. Single-cell antibiotic susceptibility. (a)** Final number of cells, counted at the bottom of each droplet at $t = 24h$, as a function of antibiotic concentration. Colors indicate the initial number of cells $i$ in the droplet at $t = 0$. The horizontal dashed line is the threshold, fixed at 15 cells, chosen to define the final droplet state as positive or negative. **(b)** Survival probability, computed using Eq (3), as a function of the initial cell number $i$ and antibiotic concentration. **(c)** The probability to produce a colony as a function of initial number of cells, for $C_{AB} = 8$ ng/mL. Bars indicate experimental measurements from counting positive cells. Dashed line shows fit according to Eq (2), assuming independent outcomes for each cell and fitting for $q$. For additional fits, see S1 Fig. **(d)** The single-cell antibiotic susceptibility is plotted for all antibiotic concentrations and all experimental replicates (points). The data is well-fitted with a two-parameter Hill function (line): $h(C_{AB}) = C_{AB}^n/(K + C_{AB}^n)$, with best fit values $n = 3.9$ and $K = 4.4$.

A threshold value to distinguish positive from negative droplets is chosen at $n_{final} = 15$ cells to separate best the two populations (colony or no colony). The number of positive and negative droplets for each condition and each initial number of cells is then determined. As a result, the probability of a droplet to contain a colony at the end-point can be plotted as a function of its initial number of bacteria $i$, and of the antibiotic concentration $C_{AB}$. We call this probability $p(i, C_{AB})$: it represents the probability to produce a colony starting from $i$ cells, under a concentration of antibiotics $C_{AB}$. The evolution of $p$ is shown as a heat-map in Fig 4b. We observe that $p(i, C_{AB})$ decreases as the antibiotic concentration increases, but that it increases with the initial number of cells in a droplet; when more cells are in a droplet initially, more antibiotics are needed to prevent the growth of a colony after an overnight incubation.

Counting the number of bacteria at initial times, and the fraction of positive droplets at the end of the experiment, enables to infer the susceptibility of a single cell to a concentration of antibiotic $C_{AB}$. This single-cell susceptibility, which we denote $q(C_{AB}) = 1 - p(1, C_{AB})$, is defined as the probability for a single cell to die (equivalently not form a colony) at concentration $C_{AB}$. If additionally all bacteria are assumed to behave independently, the probability for $i$ cells to die is $q(C_{AB})^i$. The probability for $i$ cells to form a colony is therefore the probability for at least one of them to form a colony, and we then have:

$$p(i, C_{AB}) = 1 - q(C_{AB})^i. \quad (2)$$

For each concentration of antibiotics, the probability $p(i, C_{AB})$ was estimated by counting the number $\mathcal{N}_i$ of droplets containing exactly $i$ cells at the beginning of the experiment. Among these $\mathcal{N}_i$ droplets, a number $N^+(i, C_{AB})$ droplets were positive at the end of the experiment. We then have:

$$p(i, C_{AB}) = \frac{N^+(i, C_{AB})}{\mathcal{N}_i}. \tag{3}$$

The probability $p(i, C_{AB})$ was fitted to the functional form of Eq (2) with $q(C_{AB})$ as a single fit parameter, as shown in Fig 4c (see S1 Fig for all data). The good agreement between the data and the theory confirms that the bacteria may indeed be considered as independent of each other (see in Fig 4c), at least for the low number of cells present initially at the beginning of the experiments. Then if we write $q^\star(C_{AB})$ the best fit value of $q(C_{AB})$, this value provides the best estimate of the single-cell susceptibility to drug concentration $C_{AB}$, using all experimental data at hand and assuming that all cells are independent.

The single-cell susceptibility $q^\star(C_{AB})$ is expected to take a sigmoidal shape, with a value near 0 in the absence of drugs (all bacteria form colonies), and increases non-linearly with the drug concentration until reaching a plateau at $q^\star(C_{AB}) = 1$. This is confirmed by the experimental measurements, as shown in Fig 4d. These data are arbitrarily fitted to a Hill function: $h(C_{AB}) = C_{AB}^n/(K + C_{AB}^n)$, with $K$ and $n$ being two fit parameters, providing a good match between experiments and the fitted function.

## Identifying morphology changes under antibiotic stress

This microfluidic device and image analysis pipeline is a very powerful tool to obtain quantitative information at the single-cell level as seen in Fig 4. But more than quantitative information, the anchored droplet format provides a unique ability to access qualitative data and to follow the evolution of the bacterial colonies within each droplet over time. The tracking of the droplet contents can be performed by time-lapse microscopy on the chips. Since the droplet position is invariant throughout the experiment, identifying the evolution of the morphology within each droplet can be achieved. Examples of a sample droplet contents can be seen in Fig 5 and the accompanying movies, first in the absence of antibiotics (Fig 5a) or under a sub-MIC concentration of ciprofloxacin (Fig 5b).

In the absence of antibiotics, almost all droplets display bacteria in their planktonic state, swimming in the droplets and showing the typical size and shape for *E. coli* in culture. A few

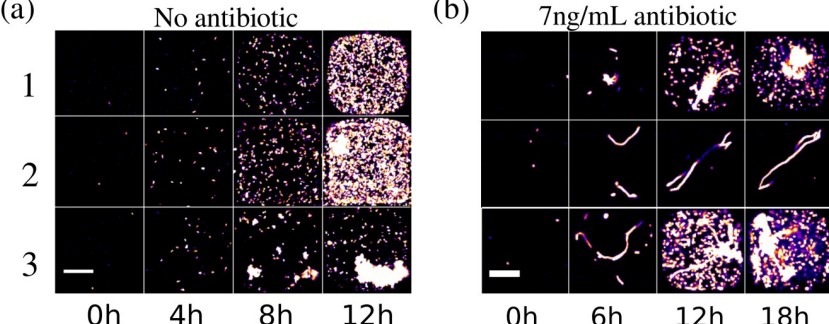

**Fig 5.** Example of bacterial growth in three independent droplets growth (**a**) without antibiotic, and (**b**) with sub-MIC concentrations of ciprofloxacin (7 ng/mL). Scale bar: 50 μm. Even at sub-MIC concentrations, the presence of ciprofloxacin leads to a filamentous bacterial morphology.

hours into the experiments and as the populations begin to grow, cells in some droplets start to adhere to each other, eventually forming clumps. The presence of these clumps could be attributed to the presence of adhesion proteins like fimbrial adhesins that may then contribute to biofilm formation of *E. coli* [31, 32].

The morphological change of bacteria in the presence of antibiotics often begins with the elongation of the cells into long filaments: this corresponds to 83% of cases in the experiments at sub-MIC antibiotic concentration (7 ng/mL). The filamentation is due to the SOS response triggered by the presence of ciprofloxacin; cell-division stops but the cell metabolism continues, leading to cell volume growth. The elongated cells can then begin to produce offspring after several hours (50% of all cases) or can arrest their growth and stay in the filamentous form (33% of all cases). This is in agreement with observations on agar plates [4].

In some cases, no elongation is found. These droplets correspond either to cells that were in a dormant state (10% of all cases is expected) [33] or to cells that suffered too much damage and died due to the antibiotics (7% of all cases) as no fluorescence can be detected. As a result of these dynamics, nearly all of the droplets that form colonies contain cells having the filamentous form as well. Nevertheless, the ability to image the contents of the droplets provides a very precise indication of the state of the cells within them. This in turn is informative about the ability of the cells to overcome the antibiotic stress.

## Discussion

As the emergence of antibiotic resistance is accelerating, it is crucial to understand the variability of antibiotic response on the single-cell level. This has motivated work using dilution methods to determine the MIC on petri dishes [5] or in liquid media [6], or using precision microscopy on agar [4]. In parallel, microfluidic methods have been used to provide better controlled conditions and statistics over a large number of individual cells [7, 12]. In recent years a flurry of droplet-based approaches has provided information showing the heteroresistance of a bacterial population [21], digital antibiotic susceptibility [22, 34], or even providing pathogen identification [25]. These droplet-based methods have shown that the encapsulation within droplets can be used to explore the progeny of individual cells as they respond to antibiotic stress. By relying on snapshots of moving droplets, these methods cannot relate the initial and final states of each droplet, nor can they identify the biological mechanisms that allow cells to overcome the antibiotic stress at sub-MIC concentrations.

In this context the device and analysis pipeline presented here combine the advantages of microscopy with those of droplet-based methods. Indeed the ability to identify the initial state of each droplet allows us to work at much higher cell numbers per droplet, which translates to similar statistics as previous papers [22] while using a much lower number of droplets in total. Moreover the ability to provide time-resolved measurements on each droplet provides unique qualitative information about the adaptation of the cells, similarly to other microscopy-based methods [4]. Finally, the streamlined experimental and analysis pipeline allows us to load and image the chip in under 30 minutes, followed by an overnight incubation and a second scan requiring only a few minutes. As a result the complete campaign for obtaining an antibiogram can be performed robustly in a few hours. In contrast, the classical (petri-dish) method requires an initial overnight incubation in a 96 well plate, followed by a second overnight incubation on 50 to 80 petri dishes and manual counting of the colonies thereafter.

Beyond the performance aspects, it is important to benchmark the microfluidic measurements against standard microbiology protocols. The comparison of MIC, MBC, and µf · MIC shows that the results in the microfluidic chip can be treated as a population-level measurement, since the µf · MIC matches the MBC when accounting for the starting inoculum size. In

addition to this, the detection of individual cells and the ability to perform a large number of single-cell assays in parallel allow us to develop a probabilistic treatment of the outcome within each droplet forming a colony after 24 hours. The probability for an individual cell to produce a colony at a given antibiotic concentration thus describes the heterogeneity of responses in the population.

Compared with other droplet-based methods, the current approach is based on analyzing a much lower number of droplets per condition (500 drops here, compared with typically $10^4$ drops [22]). However this small number of drops is partially balanced by the ability to work with a much larger Poisson parameter: Since we count the number of cells initially in each droplet, we can work with a mean number of cells above one per droplet, while other methods must work with a mean number closer to 0.1 cells per drop. The resulting total number of cells that is analyzed is therefore comparable between the different approaches. Moreover most droplet-based methods treat the biological problem as a "digital" problem, categorizing drops as either positive or negative. In contrast, the current approach provides information about cell morphology within each droplet, by relying on microscopy-based readout. While this makes the imaging and analysis of the experiments more computationally demanding, it also provides richer information about the bacterial response to antibiotic stress. In turn this information will be valuable for identifying mechanisms by which cells escape antibiotic stress.

This platform can now be used to address the single-cell response to antibiotics while screening different bacteria and molecules having different mechanisms of action. The expectation is that this screening will translate into both quantitative (values of the single cell susceptibility curve) and qualitative (shape of the cells) differences among the conditions. The ability to encapsulate tens or hundreds of cells within the droplets, in a controlled manner, will then allow the exploration of collective behaviors and non-monotonic time-evolution of the response to antibiotics [35]. The current pipeline can be combined with the results of Ref. [29] to perform more complex experiments, e.g. to modulate the antibiotic concentration in time to to recover the contents of individual drops and perform -omics measurements on them. Taken together, the different operations that can be combined into this platform constitute a major step forward in the study of antibiotic response both for scientific questions and for medical applications.

## Materials and methods

### Microfluidics and microfabrication

**Microfabrication and chip design.** To produce microfluidic chips for the experiments, a custom mold was made using 2-layer-SU-8 photoresist lithography. The bottom layer contains the channels with 2 inputs and 1 output. The upper layer contains inverted microwells of squares shapes 120x120 um and space 240 um apart. As shown in Fig 1(a), the wells are organized in 13 rows of 39 and 38 wells respectively for odds and even rows. The channel height is 30 um and the well height is 130 um.

Next, PDMS and its curing agent (PDMS SYLGARD 184, Dow Corning) are mixed at a 1:10 ratio and poured into the mold. The mold is placed in a vacuum chamber for 30 minutes to eliminate air bubbles and then cured for 3 hours at 70˚C. Once the PDMS is cured, the chip is cut off from the mold and plasma bonded (CUTE Plasma, Femto Science) to the coverslip.

**Hydrophobic treatment.** Prior to loading, the chips are surface treated with hydrophobic solution (NOVEC 1720 surface modifier/electronic grade coating 3M). To do so, two surface treatments were done after the plasma bonding by filling the chip with the hydrophobic solution and curing it for 10 minutes at 110 C. A third surface treatment is done prior to loading.

**Chip loading.** As described in Fig 1, using a syringe pump (NEMESYS), the chip is first filled through the first input with a continuous oil phase (3M Fluorinert FC40 with non-ionic surfactant RAN fluoSurf final concentration 1%) while purging the air bubbles. Then, through the second input, the continuous oil phase is replaced by the bacterial suspension. Finally, the continuous oil phase is injected again at a very low flow rate, breaking the droplets apart and leaving them locked in the wells.

## Cell culture and preparation

**Strain.** The experiments were performed using the E. coli W3110 strain (W3110 lacYZ:: mRFP-1 (JEK1037)) kindly provided by the Austin Lab [36]. JEK1037 carries a fluorescent protein (mRFP) inserted in the lac operon (lacYZ::mRFP-1). 0.5 mM IPTG was used to induce the expression of mRFP.

**Antibiotic solution.** Ciprofloxacin (Sigma-Aldrich) was solubilized in 0.1 N HCl (Sigma-Aldrich) at 25 mg/mL. The stock was then diluted with MiliQ water to 1 μg/mL. The final concentration of ciprofloxacin used was between 0 and 32 ng/mL.

**Cell culture.** From the -80 C stock, the cells were streaked on LB agar plate and incubated overnight at 37 C. The next day one isolated colony is inoculated in supplemented minimal media (MOPS with glucose final concentration 0.4%) and IPTG is added at 0.5 mM to induce the expression of the RFP. The bacterial suspension is incubated overnight at 37 C with shaking.

**Cell dilution.** In order to get one to five cells per droplet, the optical density of the solution, measured at 600nm, was calibrated. This was achieved as follows: the calibration was known by using digital counting: the chip was loaded with the diluted bacterial suspension without antibiotics, incubated at 37 C overnight, and then imaged. The empty wells were counted, and assuming a well can only be empty if no cells is loaded, the initial loading parameter λ was computed. λ is the Poisson parameter which corresponds to the mean number of cells per droplet and can be directly obtained by $-\ln(N_{(-)}/N_{total})$. Where $N_{(-)}$ is the number of negative droplets and $N_{total}$ is the total number of droplets. λ is monitored by the concentration of the bacterial suspension and directly linked to the optical density.

## Growth characterization and antibiotic susceptibility

**Growth curves.** For growth characterization cells were loaded at different dilutions (500 to 500,000 cells per well) in a 96-well plate. The plate was then placed in the plate reader (Thermo Scientific Varioskan LUX) for 24 hours at 37 C with shaking. The optical density at 600 nm (OD) and the RFP fluorescence signal (excitation 488 nm and emission 520 nm) were measured every 10 minutes.

In parallel, cells were loaded to the chip with an average of one cell per droplet. This corresponds to one cell per droplet on average. The chip was then placed under the microscope and the RFP signal was measured every 30 minutes.

**MIC and MBC.** The minimum inhibitory concentration (MIC) and minimum bactericidal concentration (MBC) were obtained to characterize the antibiotic susceptibility at the population level.

Cells are loaded in a 96 well plate at different dilutions (500 to 500,000 cells per well) and with different ciprofloxacin concentration ranging from 2 to 36 ng/mL. The plate is then incubated for 24 hours at 37 C with shaking.

The MIC is determined as the antibiotic concentration of the first negative well, i.e. the well at which the OD is the same as the OD of an empty well.

Then, the contents of the negative wells are plated on LB agar plates and incubated for 24 hours at 37 C. The number of colonies from each of these plates is counted. The MBC is

determined as the concentration where the number of colonies decrease sharply from more than a hundred cells to less than a dozen cells.

**Single-cell susceptibility.** To characterize the single-cell susceptibility to antibiotic, at least 10 microfluidic chips are fabricated and loaded per experiment: a control chip without antibiotic, and nine more with serial concentrations of ciprofloxacin (from 2 to 36 ng/mL). The bacterial suspension is prepared with the antibiotic and immediately loaded into the chip. The chip is imaged, using fluorescence microscopy, right after in order to determine the exact number of cells in each well.

Then, the chip is immersed in a container filled with Milli Q water to prevent evaporation and incubated overnight at 37 C. After incubation, a scan of each chip, measuring the RFP signal is performed, using fluorescence microscopy.

## Microscopy and image acquisition

Microscopy images are acquired using spinning disk confocal microscope (Nikon Ti2 + Yokogawa) with a 20x 0.7 NA air objective lens (Nikon Inc.) and with 2x2 pixels binning (set directly in camera properties (Hammamatsu Orca 4)). Images of the complete chip are obtained by stitching individual images with a 5% overlap. The imaging rate is optimized by acquiring first a bright-field image of the complete chip. The RFP signal is then obtained in confocal mode, using 3D stack.

For the 3D stack, using triggered NIDAQ Piezo Z, planes are acquired with a step of 1 μm, for a total penetration of 120 μm.

The total area of the device is about 14 by 4 mm, which translates to 40 kpix long dataset using 350 nm pixel size. However, some chips could be tilted due to manual bonding, which effectively increases the necessary scanning area, resulting in much bigger images.

## Image analysis

**Image registration.** First the 3D fluorescence stack is converted to 2D using maximum projection (see Fig 2b and 2e). Both channels, 2D bright-field and 2D fluorescence images, are merged together and saved as a tif stack. Then a well-labeled template image and a well-labeled mask are made and aligned to the experimental images (see Fig 2c and 2f).

Note that the protocols described above can be modified for particular situations. For instance time-lapse microscopy can be performed on the chips, in order to obtain time-resolved measurements. Similarly, confocal imaging can be used at later times in order to obtain a more precise cell count or fluorescent intensity, or to identify the morphology of the cells in particular cases. Although these cases would require small modification in the pipeline, the main bricks of the analysis discussed above can still be used in a modular fashion, without major changes in the general approach.

**Cell counting.** After image registration, every droplet is associated with the labeled area defined by the mask. Inside this mask, peak detection is performed to detect single fluorescent cells. To avoid false detection, due to noise, a preprocessing is performed as follows: First, a Gaussian filter, and subsequent peak detection using Scipy function peak_local_max with an absolute threshold of two. The number of peak per label is then recorded into a table for further processing.

**Statistical analysis.** Statistical significance of the data in Figs 1g and 3d was evaluated using the Python package "statannot". To test significance between growth rates for the Fig 1g, we pooled together measured growth rates from two timelapse experiments on the chips and three experiments on the plate reader. P-values were obtained with Welch's t-test for independent samples.

## Supporting information

**S1 Fig. Fitting susceptibility to susceptibility for different concentrations of ciprofloxacin.**
(TIFF)

**S2 Fig. Template (grayscale) + mask (blue) used for alignment and segmenting the droplets.** Scalebar 500 um.
(TIFF)

**S1 Video. This document shows a time-lapse movie of the whole chip, in the absence of antibiotic.** The complete image is stitched from individual fields that focus on smaller regions.
(MP4)

**S2 Video. This document shows a time-lapse movie of the whole chip at sub-MIC antibiotic concentration (7 ng/ml).** The complete image is stitched from individual fields that focus on smaller regions.
(MP4)

**S3 Video. This document shows a time-lapse movie of nine representative droplets at sub-MIC antibiotic concentration (7 ng/ml).** Note that the cells start by elongating before dividing within the droplets.
(MP4)

## Acknowledgments

The authors would like to thank the support of the Microfluidics and Biomaterials platform at Institut Pasteur. We also thank Erik Maikranz for useful discussions.

## Author Contributions

**Conceptualization:** Salomé Gutiérrez Ramos.

**Data curation:** Andrey Aristov, Salomé Gutiérrez Ramos.

**Formal analysis:** Andrey Aristov, Gabriel Amselem.

**Funding acquisition:** Charles N. Baroud.

**Investigation:** Lena Le Quellec, Salomé Gutiérrez Ramos, Gabriel Amselem.

**Methodology:** Lena Le Quellec, Andrey Aristov, Gabriel Amselem, Julia Bos, Zeynep Baharoglu.

**Project administration:** Didier Mazel, Charles N. Baroud.

**Resources:** Julia Bos, Zeynep Baharoglu.

**Software:** Andrey Aristov.

**Supervision:** Charles N. Baroud.

**Validation:** Andrey Aristov, Gabriel Amselem, Charles N. Baroud.

**Visualization:** Andrey Aristov.

**Writing – original draft:** Lena Le Quellec, Andrey Aristov, Gabriel Amselem, Julia Bos, Charles N. Baroud.

**Writing – review & editing:** Lena Le Quellec, Andrey Aristov, Salomé Gutiérrez Ramos, Julia Bos, Charles N. Baroud.

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
