## [Decision Letter · Decision Letter 0]

13 Mar 2024

PONE-D-24-06332Measuring single-cell susceptibility to antibiotics within monoclonal bacterial populationsPLOS ONE

Dear Dr. Aristov,

Thank you for submitting your manuscript to PLOS ONE. After careful consideration, we feel that it has merit but does not fully meet PLOS ONE’s publication criteria as it currently stands. Therefore, we invite you to submit a revised version of the manuscript that addresses the points raised during the review process.

We look forward to receiving your revised manuscript.

Kind regards,

Vinayak Singh, Ph.D.

Academic Editor

PLOS ONE

Journal Requirements:

   "ANR grant UniBAC (ANR-17-CE13-0010)"

5. Please amend your authorship list in your manuscript file to include author Charles N. Baroud.

Reviewers' comments:

Reviewer's Responses to Questions

**Comments to the Author**

1. Is the manuscript technically sound, and do the data support the conclusions?

Reviewer #1: Yes

Reviewer #2: Yes

2. Has the statistical analysis been performed appropriately and rigorously? 

Reviewer #1: Yes

Reviewer #2: I Don't Know

3. Have the authors made all data underlying the findings in their manuscript fully available?

Reviewer #1: No

Reviewer #2: Yes

4. Is the manuscript presented in an intelligible fashion and written in standard English?

Reviewer #1: Yes

Reviewer #2: Yes

5. Review Comments to the Author

Reviewer #1: The depleting pipeline of antibiotics and the spread of AMR necessitates new tools to classify and study antibiotic activity. In this manuscript, the authors use a previously developed microfluidic droplet-based platform to look at antibiotic susceptibility. Using fluorescent E. coli and ciprofloxacin as the candidate antibiotic they evaluate their platform. Initial experiments were carried out to benchmark the growth of the bacteria in the platform and compared with growth in a 96-well plate format. Image analysis pipelines were subsequently developed to quantitate the bacterial growth from 0h to 24h. Exposure to different concentrations of ciprofloxacin on different chips allows them to define microfluidic MIC which is closer to MBC by conventional assays rather than to the MIC values. They further analyze these data to derive single-cell antibiotic susceptibility profiles and also demonstrate the utility of the platform to document morphological changes upon antibiotic exposure.

As the authors themselves present in the introduction, there have been numerous studies that have achieved the goals this study set out to do. So, in terms of novelty this is not an entirely unique technique or device. But the study provides an additional toolset available to the research community to address these issues. The experiments are well-designed and the manuscript written well.

Specific comments:

- In the abstract the authors state “To date, neither bulk nor single-cell methods are able to link the heterogeneity of single-cell susceptibility to the population-scale response to antibiotics.” This is misleading, as there are several studies which have linked single-cell behavior to populations scale responses – publications from Balaban group, James Collins group, Losick group for example.

- Ln 48-49, the authors claim that the link between single-cell measurements and the classical biological measurements has never been explicitly tested. Again this is misleading as all the microfluidic papers on antibiotic susceptibility testing that the authors cite indeed compared the findings with classical biological measurements.

- It is not very convincing that this platform is any better than a conventional 96 well plate. Imaging of bacteria in a conventional 96 well plate should be able to accomplish most of the observations obtained on this platform. Especially since in any case the cells at the bottom of the droplet are being imaged. For example, in the study (Zahir T, Camacho R, Vitale R, Ruckebusch C, Hofkens J, Fauvart M, Michiels J. 2019. High-throughput time-resolved morphology screening in bacteria reveals phenotypic responses to antibiotics. Communications Biology 2:1–13.)

- The authors show that the microfluidic MIC is similar to the MBC determined by conventional method and not to the MIC. This was the observation with antibiotic ciprofloxacin (which seems to lyse the cells), would they expect a similar finding even with bacteriostatic antibiotics. It would have enriched the manuscript if they could have included analysis of an additional antibiotic (bacteriostatic).

- Have the authors accounted for the variable depletion of nutrients in the droplets due to differential seeding densities. Would this effect their observations on antibiotic activity, especially under conditions that the droplet is filled up with bacteria.

- The authors claims of looking at time evolution of phenotypes over time is not fully supported as from the movies, it appears the cells are in constant turbulence in the droplet and it would not be feasible to follow the same cell over time/ divisions.

- Considering that the platform is an additional resource in the antibiotic susceptibility field, it would be beneficial if the authors could summarize and discuss the advantages and disadvantages of their setup in comparison to other microfluidic devices in the field.

- Reference 1 seems incomplete.

- Movies S1 and S2 appear to be switched.

- Ln 31, ‘By’ should be ‘by’

Reviewer #2: The authors present an interesting piece of scientific research using a droplet-based approach to measure how single E.coli cells can expand to form colonies under antibiotic stress. The authors highlight how their current approach is built upon existing platforms and technologies, and could perhaps discuss more on the novelty of their current approach.

The authors should include a section in the Materials and Methods on their statistical analyses used in their research; the statistics used are mentioned in their legends.

Overall, the manuscript is well written but there are a few minor corrections:

There is an inconsistent use of "µ" throughout.

Figure 3b is not described in the legend.

There are a few spelling errors (lines: 206, 391, 477, Fig 1 legend).

6. PLOS authors have the option to publish the peer review history of their article (what does this mean?). If published, this will include your full peer review and any attached files.

Reviewer #1: No

Reviewer #2: No

---

## [Author Response · Author response to Decision Letter 0]

26 Apr 2024

Dear referees, 

We thank both referees for the assessment of our manuscript and their support for its publication. Below we answer their comments in detail. The comments are shown in black and our answers in blue (and also preceded with “AUTHORS' REPLY: “).

Reviewer #1: The depleting pipeline of antibiotics and the spread of AMR necessitates new tools to classify and study antibiotic activity. In this manuscript, the authors use a previously developed microfluidic droplet-based platform to look at antibiotic susceptibility. Using fluorescent E. coli and ciprofloxacin as the candidate antibiotic they evaluate their platform. Initial experiments were carried out to benchmark the growth of the bacteria in the platform and compared with growth in a 96-well plate format. Image analysis pipelines were subsequently developed to quantitate the bacterial growth from 0h to 24h. Exposure to different concentrations of ciprofloxacin on different chips allows them to define microfluidic MIC which is closer to MBC by conventional assays rather than to the MIC values. They further analyze these data to derive single-cell antibiotic susceptibility profiles and also demonstrate the utility of the platform to document morphological changes upon antibiotic exposure.

As the authors themselves present in the introduction, there have been numerous studies that have achieved the goals this study set out to do. So, in terms of novelty this is not an entirely unique technique or device. But the study provides an additional toolset available to the research community to address these issues. The experiments are well-designed and the manuscript written well.

AUTHORS' REPLY: 

We thank reviewer 1 for his/her positive remarks on our manuscript. We agree with the referee that this manuscript provides an additional approach to study the bacterial response to antibiotics, this way complementing the already published studies. This manuscript therefore shows the demonstration of the type of data that can be obtained. We are currently working towards gaining new biological insights by using these tools, which will hopefully lead to future publications.

END OF REPLY

Specific comments:

- In the abstract the authors state “To date, neither bulk nor single-cell methods are able to link the heterogeneity of single-cell susceptibility to the population-scale response to antibiotics.” This is misleading, as there are several studies which have linked single-cell behavior to populations scale responses – publications from Balaban group, James Collins group, Losick group for example.

AUTHORS' REPLY: 

The referee is right that our statement was too broad. We have now toned it down as follows: 

To date, linking the heterogeneity of single-cell susceptibility to the population-scale response to antibiotics remains challenging due to the trade-offs between the resolution and the field of view.

END OF REPLY

- Ln 48-49, the authors claim that the link between single-cell measurements and the classical biological measurements has never been explicitly tested. Again this is misleading as all the microfluidic papers on antibiotic susceptibility testing that the authors cite indeed compared the findings with classical biological measurements.

AUTHORS' REPLY: 

Again, we have replaced this sentence with a more focused statements, as follows:

 Finally, there is still a need to strengthen the link between the droplet-based measurements with the vast quantity of data obtained in traditional experiments.

END OF REPLY

- It is not very convincing that this platform is any better than a conventional 96 well plate. Imaging of bacteria in a conventional 96 well plate should be able to accomplish most of the observations obtained on this platform. Especially since in any case the cells at the bottom of the droplet are being imaged. For example, in the study (Zahir T, Camacho R, Vitale R, Ruckebusch C, Hofkens J, Fauvart M, Michiels J. 2019. High-throughput time-resolved morphology screening in bacteria reveals phenotypic responses to antibiotics. Communications Biology 2:1–13.)

AUTHORS' REPLY:

 The results presented in this manuscript should be viewed as a complement to the paper by Amselem et al., Lab. Chip. 2016. In that paper we showed how the microfluidic format allowed a wide range of operations on bacterial suspensions, in the presence or absence of antibiotics. Amselem et al. showed for example that it was possible to modify the antibiotic concentration in time, or to recover the contents of a single droplet out of more than 1000. These operations distinguish the microfluidic format from multiwell plates. That paper however failed to relate the final state of a bacterial colony to the initial state inside each drop, nor did it benchmark the microfluidic measurements against standard biological tools. These two aspects are now addressed in the current manuscript.

This aspect is now emphasized in the discussion, where we now state:

 The current pipeline can be combined with the results of Ref. [28] to perform more complex experiments, e.g. to modulate the antibiotic concentration in time to to recover the contents of individual drops and perform -omics measurements on them.

When comparing with 96 well plates, the ability to work with small volumes makes it possible to count the number of cells inside each droplet initially just after loading the chips. This counting step is critical for defining the single-cell susceptibility that is shown in fig. 4, which we think opens a range of possibilities for probabilistic analysis of cell behavior. The paper by Zahir et al. does show beautiful images of cells after being subjected to antibiotics. However counting the number of cells at the initial seeding would be practically impossible in the volume of a well of a 96 well plate, since it would be prohibitively difficult to scan the whole volume to identify a single cell in 100 µl. 

END OF REPLY

- The authors show that the microfluidic MIC is similar to the MBC determined by conventional method and not to the MIC. This was the observation with antibiotic ciprofloxacin (which seems to lyse the cells), would they expect a similar finding even with bacteriostatic antibiotics. It would have enriched the manuscript if they could have included analysis of an additional antibiotic (bacteriostatic).

AUTHORS' REPLY: 

Our paper indeed focuses on the ciprofloxacyn because the SOS response due to the DNA damage by ciprofloxacyn provokes cell elongation perfectly observable inside of the droplet. The platform is naturally suitable for other antibiotics, which is the subject of our current work where we are comparing the statistics and morphologies of cells subjected to a range of different antibiotics. However these tests require a much more in-depth study, making them out of the scope of the current paper. 

END OF REPLY

- Have the authors accounted for the variable depletion of nutrients in the droplets due to differential seeding densities. Would this effect their observations on antibiotic activity, especially under conditions that the droplet is filled up with bacteria.

AUTHORS' REPLY: 

The outcome for each droplet (whether it is positive or negative and whether it leads to filamentation) is mainly determined by the initial evolution within the drop. In all of the experiments shown here, the number of cells at this early stage is in the range of a few cells per droplet, so that crowding effects or cell-cell interactions should be negligible. 

END OF REPLY

- The authors claims of looking at time evolution of phenotypes over time is not fully supported as from the movies, it appears the cells are in constant turbulence in the droplet and it would not be feasible to follow the same cell over time/ divisions.

AUTHORS' REPLY: 

The reviewer here is pointing to one of the main differences between the current approach and “mother machine” approaches. While the mother machine provides a method to follow the progeny of a specific cell, in terms of division rate and shape, the current approach only provides statistical measurements of the progeny of a small number of cells per droplet. Indeed, since the cells move it is not possible to generate a single-cell lineage. Nevertheless the format provides a way to obtain a good estimate of the distribution of shapes at each time step of the time lapse.

We have changed the statement on “progeny of single cells” as follows:

Since the droplet position is invariant throughout the experiment, identifying the evolution of the morphology within each droplet can be achieved

END OF REPLY

- Considering that the platform is an additional resource in the antibiotic susceptibility field, it would be beneficial if the authors could summarize and discuss the advantages and disadvantages of their setup in comparison to other microfluidic devices in the field.

AUTHORS' REPLY: 

We have now added a new paragraph in the discussion section. In this paragraph we compare with droplet-based microfluidic methods, which are the most relevant comparison point. The new paragraph is cited below:

Compared with other droplet-based methods, the current approach is based on analyzing a much lower number of droplets per condition (500 drops here, compared with typically $10^4$ drops~\\cite{scheler_droplet-based_2020}). However this small number of drops is partially balanced by the ability to work with a much larger Poisson parameter: Since we count the number of cells initially in each droplet, we can work with a mean number of cells above one per droplet, while other methods must work with a mean number closer to 0.1 cells per drop. The resulting total number of cells that is analyzed is therefore comparable between the different approaches. Moreover most droplet-based methods treat the biological problem as a "digital" problem, categorizing drops as either positive or negative. In contrast, the current approach provides information about cell morphology within each droplet, by relying on microscopy-based readout. While this makes the imaging and analysis of the experiments more computationally demanding, it also provides richer information about the bacterial response to antibiotic stress. In turn this information will be valuable for identifying mechanisms by which cells escape antibiotic stress.

END OF REPLY

- Reference 1 seems incomplete.

- Movies S1 and S2 appear to be switched.

- Ln 31, ‘By’ should be ‘by’

AUTHORS' REPLY: 

Thank you for your careful reading of the manuscript. We have fixed these issues and replaced Ref. 1 by a more relevant one.

END OF REPLY

Reviewer #2: The authors present an interesting piece of scientific research using a droplet-based approach to measure how single E.coli cells can expand to form colonies under antibiotic stress. The authors highlight how their current approach is built upon existing platforms and technologies, and could perhaps discuss more on the novelty of their current approach.

THE AUTHORS' REPLY: We thank reviewer 2 for his/her evaluation of our article and support for publication. The reviewer’s comment is similar to the one by reviewer 1. As a result we have now added a new paragraph in the discussion section where the current approach is compared with other droplet-based methods. The paragraph is copied below:

Compared with other droplet-based methods, the current approach is based on analyzing a much lower number of droplets per condition (500 drops here, compared with typically $10^4$ drops~\\cite{scheler_droplet-based_2020}). However this small number of drops is partially balanced by the ability to work with a much larger Poisson parameter: Since we count the number of cells initially in each droplet, we can work with a mean number of cells above one per droplet, while other methods must work with a mean number closer to 0.1 cells per drop. The resulting total number of cells that is analyzed is therefore comparable between the different approaches. Moreover most droplet-based methods treat the biological problem as a "digital" problem, categorizing drops as either positive or negative. In contrast, the current approach provides information about cell morphology within each droplet, by relying on microscopy-based readout. While this makes the imaging and analysis of the experiments more computationally demanding, it also provides richer information about the bacterial response to antibiotic stress. In turn this information will be valuable for identifying mechanisms by which cells escape antibiotic stress.

END OF REPLY

The authors should include a section in the Materials and Methods on their statistical analyses used in their research; the statistics used are mentioned in their legends.

AUTHORS' REPLY: 

Thank you for your careful reading of the manuscript. We have added the missing section.

END OF REPLY

Overall, the manuscript is well written but there are a few minor corrections:

There is an inconsistent use of "µ" throughout.

Figure 3b is not described in the legend.

There are a few spelling errors (lines: 206, 391, 477, Fig 1 legend).

AUTHORS' REPLY:

Thank you for your careful reading of the manuscript. We have fixed these issues.

END OF REPLY

---

## [Editor Report · Decision Letter 1]

30 Apr 2024

Measuring single-cell susceptibility to antibiotics within monoclonal bacterial populations

PONE-D-24-06332R1

Dear Dr. Andrey Aristov,

We’re pleased to inform you that your manuscript has been judged scientifically suitable for publication and is accepted for publication.

An invoice will be generated. Please note, if your institution has a publishing partnership with PLOS and your article meets the relevant criteria, all or part of your publication costs will be covered. Please make sure your user information is up-to-date by logging into Editorial Manager at Editorial Manager® and clicking the ‘Update My Information' link at the top of the page. If you have any questions relating to publication charges, please contact our Author Billing department directly at authorbilling@plos.org.

Kind regards,

Vinayak Singh, Ph.D.

Academic Editor

PLOS ONE
---

## [Editor Report · Acceptance letter]

16 May 2024

PONE-D-24-06332R1 

PLOS ONE

Dear Dr. Aristov, 

I'm pleased to inform you that your manuscript has been deemed suitable for publication in PLOS ONE. Congratulations! Your manuscript is now being handed over to our production team.

Kind regards, 

on behalf of

Dr. Vinayak Singh 

Academic Editor

PLOS ONE